

# Sound production in the Meagre, *Argyrosomus regius* (Asso, 1801): intraspecific variability associated with size, sex and context

Beatriz P. Pereira[1,2], Manuel Vieira[2,3], Pedro Pousão-Ferreira[4], Ana Candeias-Mendes[4], Marisa Barata[4], Paulo J. Fonseca[3] and Maria Clara P. Amorim[2,5]

[1] Faculdade de Ciências e Tecnologias, Universidade do Algarve, Faro, Portugal
[2] MARE—Marine and Environmental Sciences Centre, ISPA - Instituto Universitário, Lisboa, Portugal
[3] Departamento de Biologia Animal and cE3c—Centre for Ecology, Evolution and Environmental Changes, Faculdade de Ciências, Universidade de Lisboa, Lisboa, Portugal
[4] Instituto Portugês do Mar e da Atmosfera, Olhão, Portugal
[5] Departamento de Biologia Animal, Faculdade de Ciências, Universidade de Lisboa, Lisboa, Portugal

Corresponding author
Beatriz P. Pereira,
beatrizpalinhospereira@gmail.com

## ABSTRACT

**Background**. Many fish taxa produce sound in voluntary and in disturbance contexts but information on the full acoustic repertoire is lacking for most species. Yet, this knowledge is critical to enable monitoring fish populations in nature through acoustic monitoring.

**Methods**. In this study we characterized the sounds emitted during disturbance and voluntary contexts by juvenile and adult meagre, *Argyrosomus regius*, in laboratory conditions. Breeding sounds produced by captive adults were also compared with meagre sounds registered in the Tagus estuary (Lisbon, Portugal) from unseen fish during the breeding season.

**Results**. The present dataset demonstrates for the first time that in this species dominant frequency is inversely related to fish size, and that sounds vary according to sex, context and age. Sounds from captive breeding adults were similar to sounds recorded in the field.

**Discussion**. Our findings indicate that *A. regius* sound features carry information about size, sex, age and motivation. This variability could potentially be used to identify meagre in the field and to infer about ontogenetic phase (i.e., juveniles vs. adults, and variation with size) and motivation (e.g., spawning). Future studies should confirm sex differences and ascertain the influence of water temperature on acoustic features.

## INTRODUCTION

Numerous fish species produce sounds for social communication and when disturbed (*Radford, Kerridge & Simpson, 2014*), through several sound production mechanisms (*Ladich & Fine, 2006*; *Fine & Parmentier, 2015*). Sonic mechanisms are very diverse within

this taxon, with fish sounds varying between different species and in some cases with size and sex as well (*Myrberg Jr, Ha & Shamblott, 1993*; *Connaughton, Taylor & Fine, 2000*; *Fine & Parmentier, 2015*). Acoustic variability is mainly related to behavioural functions, such as courtship, spawning, agonistic behaviour, competitive feeding, and disturbance (*Amorim, 2006*; *Ladich & Myrberg, 2006*). This inter- and intra-specific variability of fish sounds is useful to ascertain the presence, identity and activity of fish in nature (e.g., *Borie et al., 2014*; *Erisman & Rowell, 2017*).

Sciaenidae, collectively known as the croakers and drummers, is one of the largest families of vocal fishes (*Chao, 1986*), known for the conspicuous chorusing behaviour associated with the breeding season (e.g., *Luczkovich et al., 1999*; *Parsons, McCauley & Mackie, 2013*). Representatives of this family produce swimbladder-related sounds by contracting specialized extrinsic sonic muscles against the swimbladder wall (reviewed in *Fine & Parmentier, 2015*). In most sciaenid species, sound-producing ability is restricted to males (*Tower, 1908*; *Fish & Mowbray, 1970*; *Hill, Fine & Musick, 1987*; *Connaughton & Taylor, 1995*). However, in some species, such as the black drum (*Pogonias cromis*), white croaker (*Genyonemus lineatus*), and meagre (*Argyrosomus regius*), both males and females possess sonic muscles (*Fish & Mowbray, 1970*; *Takemura, 1978*; *Lagardère & Mariani, 2006*; *Tellechea et al., 2010*).

Passive acoustic monitoring of sciaenids has been used to assess spatial and temporal patterns of fish reproduction by using sounds related with spawning (*Fish & Cummings, 1972*; *Takemura, 1978*; *Saucier & Baltz, 1993*; *Lagardère & Mariani, 2006*; *Ueng, Huang & Mok, 2007*; *Parsons et al., 2009*; *Montie et al., 2016*; *Parmentier et al., 2018*). However, although several studies on breeding vocalizations have been published for a number of sciaenid species, detailed descriptions of sound parameters and their size-dependent variations are scarce and often restricted to disturbance sounds (*Connaughton, Taylor & Fine, 2000*; *Tellechea et al., 2010*). For example, *Connaughton, Taylor & Fine (2000)* found an inverse relation between dominant frequency of disturbance sounds and fish size. These authors showed that for weakfish, *Cynoscion regalis*, throughout a range of 11 cm of total length (25–36 cm) the dominant frequency decreased by 81 Hz from 560 to 479 Hz. Nevertheless, similarities between breeding (e.g., advertisement) and disturbance sounds, namely in dominant frequency, suggest that the intraspecific variability of disturbance sounds may bear a parallel with that of advertisement sounds. Sex-related differences of vocalizations are also rarely reported since for the majority of the studied sciaenid species, sonic muscles are absent in females. Female and male sounds were described for *Micropogonias furnieri*, *Pogonias cromis*, *Cynoscion regalis* (*Tellechea et al., 2010*; *Tellechea et al., 2011*; *Tellechea & Norbis, 2012*), and *Argyrosomus japonicus* (*Ueng, Huang & Mok, 2007*). *Tellechea et al. (2011)* showed that for black drum (*Pogonias cromis*) only males produced advertisement sounds, while both sexes emitted sounds when disturbed. *Ueng, Huang & Mok (2007)* observed that in Japanese croaker (*Argyrosomus japonicus*) the advertisement sounds of the male and female differed; females generated significantly more pulses per call, and their sounds had a longer call duration, a shorter pulse period, and a lower dominant frequency than those of males.

The meagre, *Argyrosomus regius* (Asso, 1801), is a vocal sciaenid widely distributed along the Atlantic coast of Europe and Africa, and in the Mediterranean (*Chao, 1986*). It is a species with a high commercial value, being farmed in several countries since the 1990s (*Monfort, 2010*; *FAO, 2011*). While advertisement sounds produced during spawning aggregations and the sexual dimorphism of sonic muscles have been characterised in this species (*Lagardère & Mariani, 2006*; *Vieira et al., 2019*), there is still a lack of information on their wider vocal repertoire, including whether females make sounds, and the variability of sound production associated with social context (disturbance vs. reproductive) or ontogenetic phase (vocal repertoire of juvenile vs. adult, and variation with size). Such knowledge could be useful to identify juveniles and adult meagre in the field, e.g., allowing recognition of young fish shoals and reproduction areas.

The present study aimed to describe disturbance and voluntary sounds produced by juvenile and adult meagre to examine intraspecific variability of the acoustic signals associated with context, size and sex. In addition, we aimed to compare sounds registered in an aquaculture setup with field recordings. Our data suggests that it might be possible to monitor the meagre in the field using passive acoustics, and that intraspecific variability in the acoustic signals renders the opportunity to discriminate ontogenetic groups and reproduction stages.

## MATERIALS AND METHODS

### Captive fish

Sound recordings were obtained from a group of adult breeders and juvenile meagre, *A. regius*, reared at the aquaculture facilities of Instituto Português do Mar e da Atmosfera —Estação Piloto de Piscicultura de Olhão (IPMA –EPPO), Portugal (37°02′ N, 7°49′ W). IPMA is an institution certified to perform experimental work with animals. It has an authorization according to EU legislation for EPPO to breed, use and supply aquatic animals for scientific experimental work provided by DGAV—the Portuguese National Authority for Animal Health—(DGAV reference 0421/000/000/2018). All fish were housed in aquaculture tanks with continuous filtered water supply, under controlled water temperature (ranging from 14 °C to 23 °C, measured once a day), controlled pH ($8 \pm 0.4$), salinity ($37 \pm 1.0$ psu) and oxygen level close to saturation ($80 \pm 7.6\%$). Photoperiod conditions were set to mimic natural regimes (10L/14D hours, with low light intensity at dawn and dusk). Juveniles' photoperiod was larger; these individuals were provided with natural sunrise light, and artificial light prolonged the photoperiod until 23:00 all year round.

The studied adult breeders ($n = 10$) 6 and 9 years old, were reared in indoor concrete parallelepipedic tanks (3 m$^2$ area, 1.2 m deep), and were the offspring of wild individuals hosted in the aquaculture facilities. Adult meagre exhibited an average total length (TL) of 87 cm ranging from 69–102 cm, and a 4:6 (M:F) sex ratio. These individuals were fed inert semi-moist feeds several times a day. Subject juveniles were reared in two indoor 9000 L fiberglass circular tanks (3 m diameter, 1.6 m deep). One tank housed 1 and 2 year-old juveniles ($n = 208$) ranging in TL from approximately 30 to 50 cm, respectively. The other
tank reared 2.5 month-old juveniles ($n = 4,000$), with an average TL of ca. 9 cm. Note that meagre attains maturity at ca. 86 cm TL but only shows signs of spawning activity at ca. 100 cm TL in the wild (*Prista et al., 2014*), while in aquaculture conditions it can mature at shorter lengths (length at 50% maturity at ca. 50 cm and around 3 years of age; *Gil et al., 2013*). The studied non-breeder fish are thus juveniles due to their size and age (TL < 50 cm; age $\leq$ 2 years) and because they never showed any signs of sexual activity.

### Recording of vocalizations in captive fish
#### Ontogenetic and sex variation of disturbance sounds
Vocalizations of juveniles with different size were individually recorded at ca. 20 °C water temperature in May 2018. Two 200 L plastic circular containers were filled with water from the rearing tanks under continuous aeration. A group of 20 juveniles with an average TL of 35 cm, ranging from 30–50 cm in TL, was transferred (10 at a time) with a hand-net to one of the 200 L containers. A single individual was then rapidly measured for total length and weight and transferred with a plastic sleeve bag to the second 200 L container where recordings were carried out (Fig. 1). This fish was kept inside the sleeve and sometimes stimulated by pressing the caudal peduncle while disturbance sounds were recorded for 3 min with a High Tech 94 SSQ hydrophone (High Tech Inc., Gulfport, MS, USA; sensitivity of −165 dB re 1 V/μPa, frequency response up to 6 kHz within ± 1 dB), placed at approximately 10 cm from the fish's abdomen, and connected to a Tascam DR-40 Portable Digital Recorder (44.1 kHz sampling rate, 16 bit; TEAC, Europe Gmbh).

To check for sound production in smaller fish, similar recordings of disturbance sounds were conducted with a group of 10 randomly selected 2.5 month-old juveniles with an average TL of ca. 9 cm. The recording protocol was similar to the one above, except that fish were transferred to a 10 L glass beaker with water from the original tank also at approximately 20 °C. As before, disturbance sounds were recorded individually. Furthermore, one individual with 10 cm TL was transported to the laboratory facilities at University of Algarve, for dissection to inspect for sound-producing muscles. For this purpose, the fish was euthanized with an overdose of MS222 (tricaine methane sulfonate solution buffered with sodium bicarbonate to a neutral pH; Pharmaq, Norway).

A similar experimental design was carried out in July 2018 for the adult breeders (see above). However, prior to sampling, the water level in the indoor tank was lowered by 1 m and all fish (fasted for 24 h) were anaesthetized with 40 ppm of 2-phenoxyethanol (2-PE) to reduce stress and prevent fish from jumping and harming themselves while being captured. Following the juveniles' recordings protocol, two 200 L containers filled with aerated seawater from the rearing tanks and without anaesthetic, were used. Water temperature was the same as in juveniles (~20 °C) to rule out differences in the sounds due to temperature-dependent effects. After 20 min, anaesthetized individuals were safely captured with a plastic sleeve bag and transferred in groups of 3 to the first 200 L container, where fish were allowed to recover for approximately 15 min after which they showed no signs of anaesthesia. Then, each breeder was individually identified by chip reading and transferred with a sleeve to the second 200 L container where recordings were carried out (Fig. 1). *Barata et al. (2016)* tested the efficiency of 2-phenoxyethanol (2-PE) in juveniles

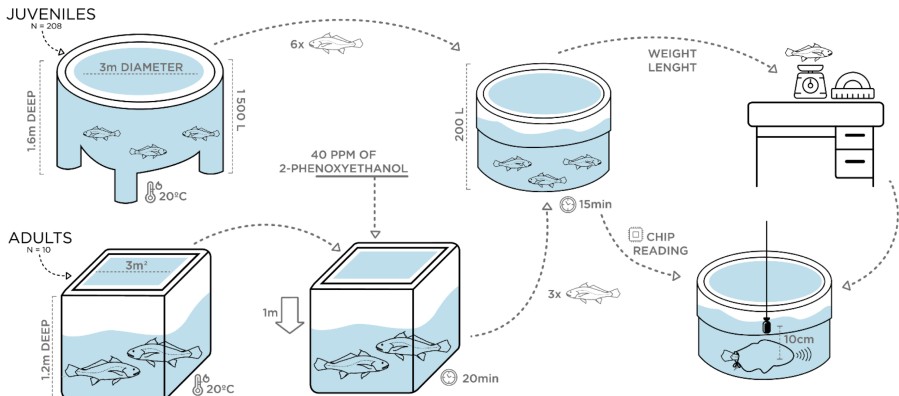

**Figure 1** Schematic figure depicting the experimental setup for the assessment of ontogenetic and sex variation of disturbance sounds.

of this species at IPMA, based on the stages of anaesthesia described by *Iwama, McGeer & Pawluk (1989)*: total loss of equilibrium (Stage AI), loss of gross body movements (Stage AII), and cessation or imperceptible opercular movements (Stage AIII). According to the study, after being exposed for 10 min to 100 mg/L of 2-PE, meagre juveniles lost equilibrium (AI) but continued to exhibit locomotion and opercular movements and took approximately 4 min do recover. Hence, we are confident that adult breeders were allowed enough time to recover. Biometric samplings were conducted a few days later, with breeders fasted for 48 h being submitted to a higher dose of anaesthetic (200 ppm). In addition to total length and weight measurements, sex determination of individuals was attempted. Since external sexual dimorphism is not present in meagre, sex identification was carried out overwater by carefully turning the fish upside down and gently pressing the abdomen to check if sperm was released through the genital papilla. In the case of sperm release, individuals were considered males. Otherwise, a flexible polystyrene catheter was inserted in the genital papilla to extract gonadal tissue and identify the fish as female. However, except for two females, gonadal tissue extraction was not succeeded, and female sex was attributed to individuals that in various samples over the previous 3 years never released milt when their abdomen was pressed.

### Ontogenetic variation of sounds made during voluntary contexts

On May 2018, recordings of juvenile individuals (on the tank with individuals ranging from ca. 30–50 cm; see above), were conducted for 2 h using a Tascam DR-40 Portable Digital Recorder connected to a High Tech 94 SSQ hydrophone vertically positioned at the centre of the tank at approximately 30 cm from the bottom. Water temperature in the tank was 19.8 °C. Sounds of adult individuals ($n = 10$; see above) with an average total length (TL) of 87 cm ranging 69–102 cm, and a 4:6 (M:F) sex ratio, were recorded during the spawning season, on July 2018. For this purpose, a High Tech 94 SSQ hydrophone was vertically positioned at the centre of the tank at approximately 30 cm from the bottom and connected to a stand-alone 16 channel datalogger (LGR –5325, Measurement Computing

Corp, Norton Ma USA; 12 kHz sampling rate 16 bit, $\pm 1$ V range). Water temperature in the tank was 21.3 °C. Note that the identity of the calling fish was not known in these recordings, both in juvenile and in adult fish.

## Vocalizations recordings in the field

Five-day round-the-clock recordings of breeding sounds were collected in May and June 2018, in the Tagus estuary (Air Force base 6, Montijo, Portugal; 38°42′N, 8°58′W). Sound recordings were obtained from wild adult breeders with unknow size, sex ratio and group size. For this purpose, a High Tech 94 SSQ hydrophone was anchored at about 20 cm from the bottom to a stainless-steel holder projecting from a concrete base where the cable was attached to minimise current-induced hydrodynamic noise. The signal from the hydrophone was recorded by a 16 channel stand-alone data logger. Diurnal temperature variation was highly influenced by the tide, with a mean of 20.2 °C and a range between 17.1–21.8 °C. Water depth varied approximately between 3–6 m, depending on tide.

## Sound analysis

Sounds were edited with Adobe Audition 3.0 (Adobe Systems Inc., CA, USA) and analysed with Raven 1.5 (The Cornell Lab of Ornithology, NY, Ithaca). Only the sounds with a good signal-to-noise ratio were used in the analyses. The following temporal parameters were measured from disturbance and voluntary sounds (Fig. 2): sound duration (ms), as the time from the onset of the first pulse to the offset of the last pulse; number of pulses; pulse period (ms), as the average time between the peaks of two consecutive pulses in a sound. Pulse period was obtained by dividing the duration of the sound by the number of pulses minus 1. The measured spectral parameters were: peak (or dominant) frequency (Hz), the frequency with the highest energy in the sound; 1st and 3rd Quartile Frequencies (Q1 and Q3 frequency, Hz), the frequencies that divide the selection into two frequency intervals containing 25% and 75% of the energy in the selection. Temporal parameters were measured from oscillograms while frequency parameters were measured from power spectra (6 kHz, FFT size 1,024 points, Hamming window, 50% time overlap) with Raven 1.5 custom tools.

To examine ontogenetic and sex variation of disturbance sounds a total of 600 sounds were randomly selected and manually analysed (20 sounds per individual for each of the 20 juveniles and 10 adults recorded). To study ontogenetic variation of sounds made during voluntary context in captivity, a total of 100 sounds (50 for juveniles and 50 for adults) were analysed. The analysed juvenile sounds were produced in May 15 around 14 h while the sounds analysed for adults were produced in July 7 between 18 h and 23 h. In the case of adults, the chosen time window corresponds to the diel vocal activity period exhibited by this species (*Vieira et al., 2019*). The selected sampling day was the one in which water temperature was the most similar to disturbance recordings.

To compare breeding sounds produced in captivity with those produced in the field an additional 20 voluntary sounds registered in captivity during the spawning season were considered, to increase sample size (a total of 70 sounds). Fifty sounds emitted during the spawning season in the Tagus estuary were analysed. These sounds were selected from field

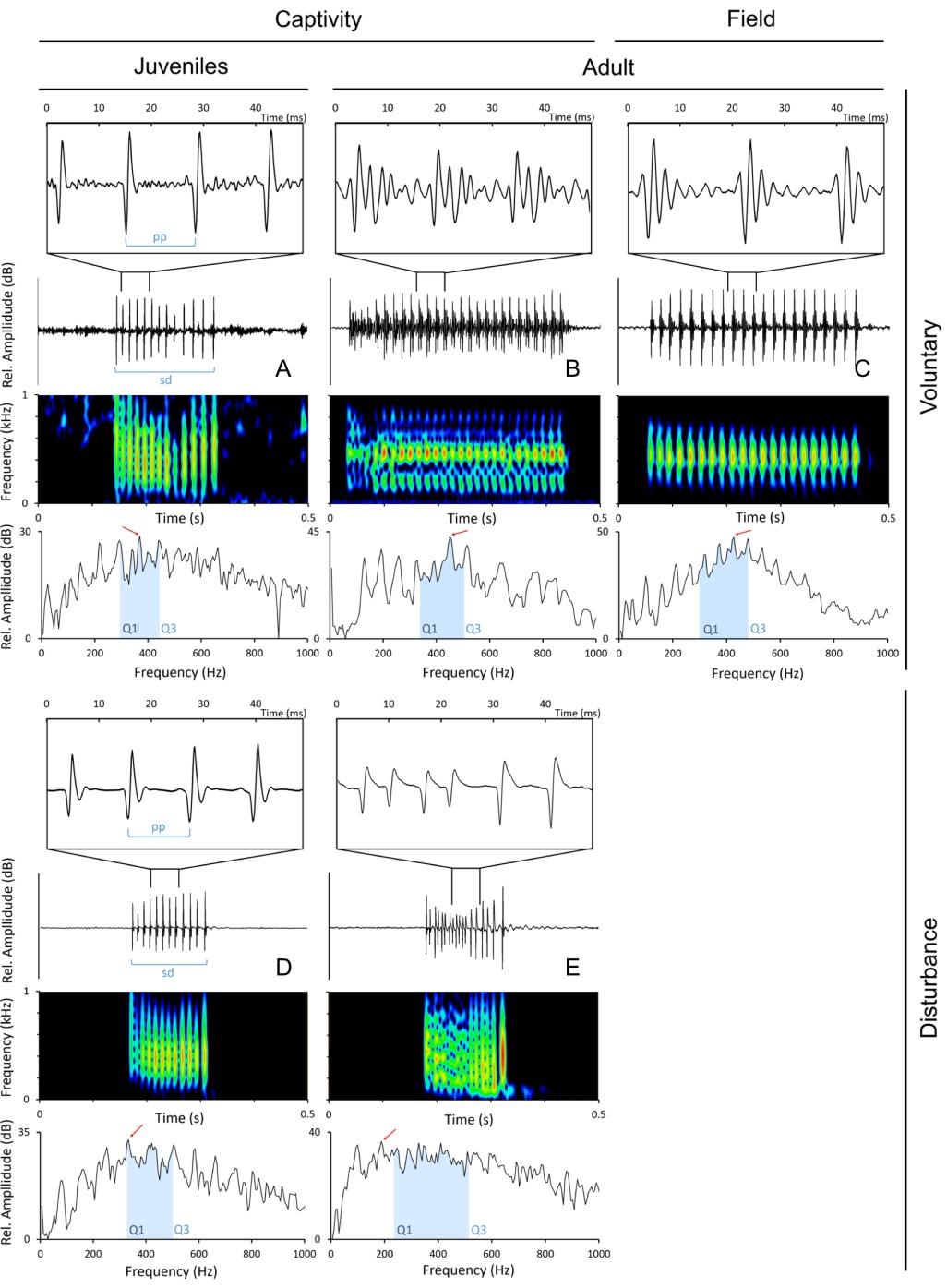

**Figure 2** **Oscillograms, spectrograms and power spectra of *Argyrosomus regius* sounds, representing some of the measured acoustic parameters.** Panels depict oscillograms (and a fragment of the oscillogram), sonograms and power spectra of (A) juvenile meagre's voluntary sound, (B) adult meagre's voluntary sound, (C) voluntary sound recorded in the field during the breeding season, (D) juvenile meagre's disturbance sound, (E) adult meagre's disturbance sound. Pulse period (pp), sound duration (sd), peak frequency (red arrow), Q1 and Q3 frequencies (blue selection; frequency intervals containing 25% and 75% of the energy in the spectrum). Spectrum and spectrogram configuration: Sampling frequency, 6 kHz; FFT size, 1,024; window points: 96; window type, Hanning; overlap samples per frame, 50%.

recordings in periods during which water temperature was similar (∼20 °C) to the one registered in captivity (∼21 °C) to minimise differences due to temperature. When meagre sounds were produced in dense choruses individual sounds could not be distinguished in field recordings. Hence, only individual sound emissions that occurred before and after the main chorus were analysed.

## Statistical analysis

Statistical analysis was conducted using the software Statistica (version 13, TIBCO Software inc, Palo Alto, CA, USA).

To assess the influence of fish total length (TL) on all acoustic variables of disturbance sounds, Pearson correlation tests were used. Data in these tests considered mean values per fish ($n = 30$, 20 sounds per fish, total of 600 sounds) for each acoustic variable. Because frequency parameters were inter-correlated ($r = 0.57–0.73$, $p < 0.05$) the effects of context, ontogenetic phase and sex were only investigated for peak frequency, the parameter that showed the highest correlation coefficient with fish TL.

To assess differences in sound features between contexts (disturbance and voluntary) and ontogenetic groups (juveniles and adults), two-way ANOVAs were conducted considering 50 sounds per context for each ontogenetic group/context. In these analyses, data concerning disturbance sounds were restricted to 50 randomly selected sounds from the whole data set to avoid large imbalances between factor level sample sizes. To meet the ANOVA assumptions the reciprocal transformation ($x' = 1/x$) was used for sound duration and number of pulses and the quadratic ($x' = x^2$) for pulse period. Post-hoc Tukey HSD tests were used to assess pairwise differences. Non-parametric two-way ANOVA was carried out for peak frequency, following (*Marôco, 2018*), since homogeneity of variances was not met even after performing the usual data transformations. To further explore fine differences in disturbance and voluntary sounds in juveniles and adults, plots were made between the different acoustic parameters and number of pulses using all analysed sounds. Differences in slopes in the relation between sound duration and number of pulses between voluntary and disturbance sounds were examined with an ANCOVA (homogeneity-of-slope model). The reciprocal transformation ($x' = 1/x$) was used for sound duration to stabilize the error variances.

To examine sex-related differences of disturbance call characteristics, an ANCOVA was performed using sex as a factor and TL as the covariate. As TL had a non-significant effect, it was removed from the analysis. Male and female sound characteristics were then compared with Student's $t$-tests . The data used in this analysis were mean values of 20 sounds for each of the 4 males and 6 females. Data met the assumptions of normality and homogeneity of variances.

Voluntary sounds recorded in captivity (total of 70 sounds) were compared with meagre sounds recorded in the field (total of 50 sounds) with ANCOVA using water temperature as the covariate. Because temperature effect was only significant for pulse period the remaining variables were compared with Student's $t$-tests . The reciprocal transformation ($x' = 1/x$) was used for number of pulses. Data met the assumptions of normality and homogeneity of variances.
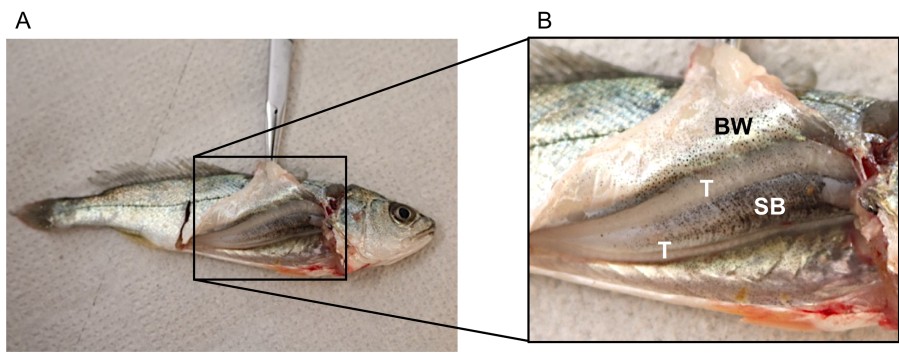

**Figure 3 Dissected juvenile meagre with a total length of ca. 9 cm depicting the sonic apparatus, lacking the extrinsic sonic muscles.** (A) Photography of juvenile (TL = 9 cm) with open body cavity. (B) Enlargement of the same picture depicting raised body wall (BW), anterior part of the swimbladder (SB) and testicles (T). No distress sounds were registered from juveniles of this size, but distress and social sounds were registered from the remaining studied juvenile fish (TL > 30 cm).

**Table 1 Relation between total length (cm) and sound features (Pearson correlation) for juvenile and adult meagre (*n* = 30 fish, 20 sounds per fish).**

| Sound features | *r* | *p*-value |
|---|---|---|
| Sound duration (ms) | 0.45 | 0.012 |
| Number of pulses | 0.63 | <0.001 |
| Pulse period (ms) | −0.59 | <0.001 |
| Peak frequency (Hz) | −0.57 | 0.001 |
| Q1 frequency (Hz) | −0.14 | 0.43 |
| Q3 frequency (Hz) | −0.54 | 0.002 |

## RESULTS

### Ontogenetic and context variation of sounds

No disturbance sounds were registered from the small juveniles of ca. 9 cm. Dissection of one individual showed that it was a male (testes were visible) but sonic muscles were not visible (Fig. 3). Figure 3 depicts the swimbladder, consisting of a single chamber, located between the viscera and the vertebral column. Disturbance and voluntary sounds were registered from the remaining studied fish (TL > 30 cm).

All acoustic features of disturbance sounds were significantly correlated with total length with the exception of Q1 frequency (Table 1, Fig. 4). Sound duration and number of pulses increased with fish size while pulse period, peak frequency and Q3 frequency decreased. On average, across a range of 71 cm in total length (31–102 cm) sound duration increased 9 ms from 126 to 135 ms, the number of pulses increased by 3 from 12 to 15, pulse period decreased by 2 ms from 11 ms to 9 ms and peak frequency decreased by 79 Hz from 340 Hz to 261 Hz.

The context of sound production had a significant effect on all sound features (Tables 2 and 3; Fig. 5). Sounds emitted in a voluntary context were longer, presented a higher number of pulses, longer pulse period, and higher peak frequency than disturbance sounds

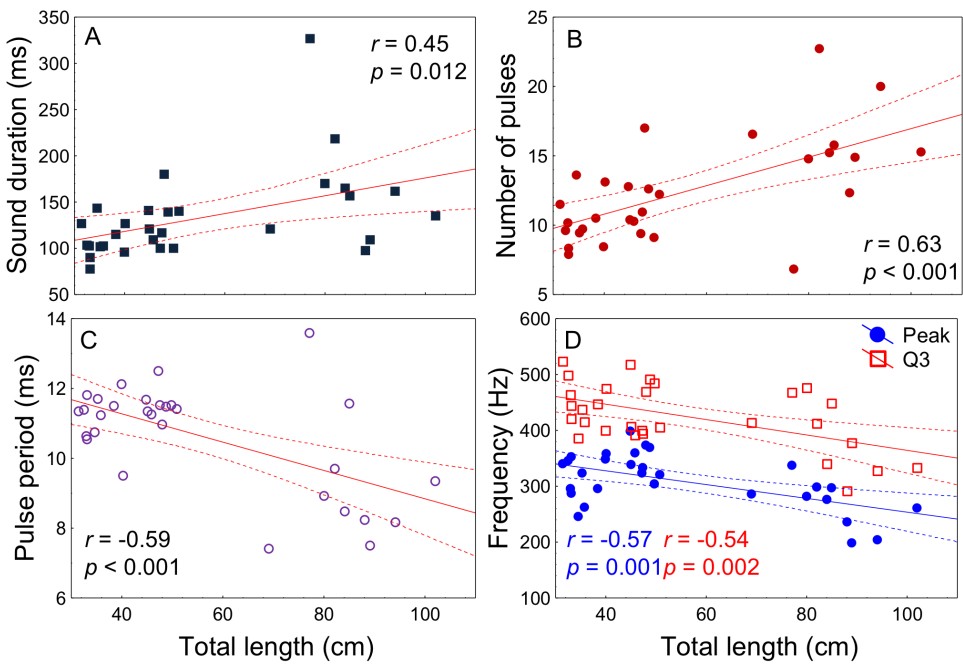

**Figure 4** **Relation of fish total length to meagre disturbance sounds parameters.** Relation between total length (cm) and (A) sound duration (ms), (B) number of pulses, (C) pulse period (ms), (D) peak and Q3 frequencies (Hz). Fish ranged from 31 to 102 cm in total length ($n = 30$) and all recordings were made at 20 °C. Regression lines and 95% confidence intervals are shown. $r$ and $p$-values are from Pearson correlation tests.

**Table 2** **Effects of context and ontogenetic phase on sound features.** Results refer to parametric two-way ANOVAs in the case of sound duration, number of pulses and pulse period following data transformations and to the equivalent non-parametric test for peak frequency. Fifty sounds were considered per ontogenetic class and context (see Materials and Methods for details).

| Factors | Sound duration (ms) | | | Number of pulses | | | Pulse period (ms) | | | Peak frequency (Hz) | | |
|---|---|---|---|---|---|---|---|---|---|---|---|---|
| | df | F | *p*-value | df | F | *p*-value | df | F | *p*-value | df | H | *p*-value |
| Context | 1,194 | 61.08 | <0.001 | 1,189 | 21.21 | <0.001 | 1,189 | 155.51 | <0.001 | 1 | 83.21 | <0.001 |
| Ontogenetic phase | 1,194 | 9.62 | 0.002 | 1,189 | 18.16 | <0.001 | 1,189 | 0.67 | 0.41 | 1 | 1.45 | 0.23 |
| Interaction | 1,194 | 0.02 | 0.88 | 1,189 | 0.19 | 0.67 | 1,189 | 3.35 | 0.07 | 1 | 3.50 | 0.06 |

in both ontogenetic groups. There was a significant effect of the ontogenetic phase on sound duration and number of pulses which increased from juveniles to adults. However, post-hoc Tukey HSD tests showed no significant differences in sound duration between juveniles and adults either for disturbance ($p = 0.14$) or for voluntary sounds ($p = 0.15$). The number of pulses increased in adults both for disturbance sounds (Tukey HSD test, $p < 0.01$) and for voluntary sounds (Tukey HSD test, $p < 0.05$).

Disturbance sounds consisted of pulse trains and presented more amplitude-modulation than voluntary sounds, with $16 \pm 3$ (mean $\pm$ SD) pulses in adult males, $15 \pm 6$ pulses in adult females and $11 \pm 4$ pulses in juveniles (Fig. 2, Table 2). The pulses of disturbance sounds were typically formed by pulses with a single major cycle (Fig. 1). The voluntary

Pereira et al. (2020), *PeerJ*, DOI 10.7717/peerj.8559

**Table 3    Sound duration, number of pulses, pulse period and peak frequency of sounds produced by meagre per ontogenetic group, sex and context.**

| | | | Sound duration (ms) | | Number of pulses | | Pulse period (ms) | | Peak frequency (Hz) | | |
|---|---|---|---|---|---|---|---|---|---|---|---|
| | | | Mean ± SD | Range | Mean ± SD | Range | Mean ± SD | Range | Mean ± SD | Range | *n* |
| **Disturbance** | | | | | | | | | | | |
| | **Adults** | **Female** (*n* = 6) | 133 ± 29 | 97–170 | 16 ± 3 | 12–21 | 8 ± 1 | 7–9 | 243 ± 41 | 192–286 | 120 |
| | | **Male** (*n* = 4) | 220 ± 85 | 156–341 | 15 ± 6 | 7–22 | 11 ± 2 | 8–13 | 305 ± 23 | 276–329 | 80 |
| | **Juveniles** | (*n* = 20) | 117 ± 46 | 15–322 | 11 ± 4 | 2–28 | 11 ± 2 | 7–22 | 329 ± 63 | 106–481 | 400 |
| **Voluntary** | | | | | | | | | | | |
| | **Adults** | **Captivity** | 626 ± 407 | 59–1,839 | 40 ± 23 | 4–116 | 16 ± 2 | 6–22 | 384 ± 110 | 123–598 | 70 |
| | | **Field** | 574 ± 304 | 68–1,268 | 28 ± 14 | 4–67 | 21 ± 3 | 8–25 | 363 ± 50 | 281–457 | 50 |
| | **Juveniles** | | 231 ± 101 | 83–554 | 16 ± 6 | 7–35 | 15 ± 1 | 13–18 | 424 ± 68 | 223–510 | 50 |

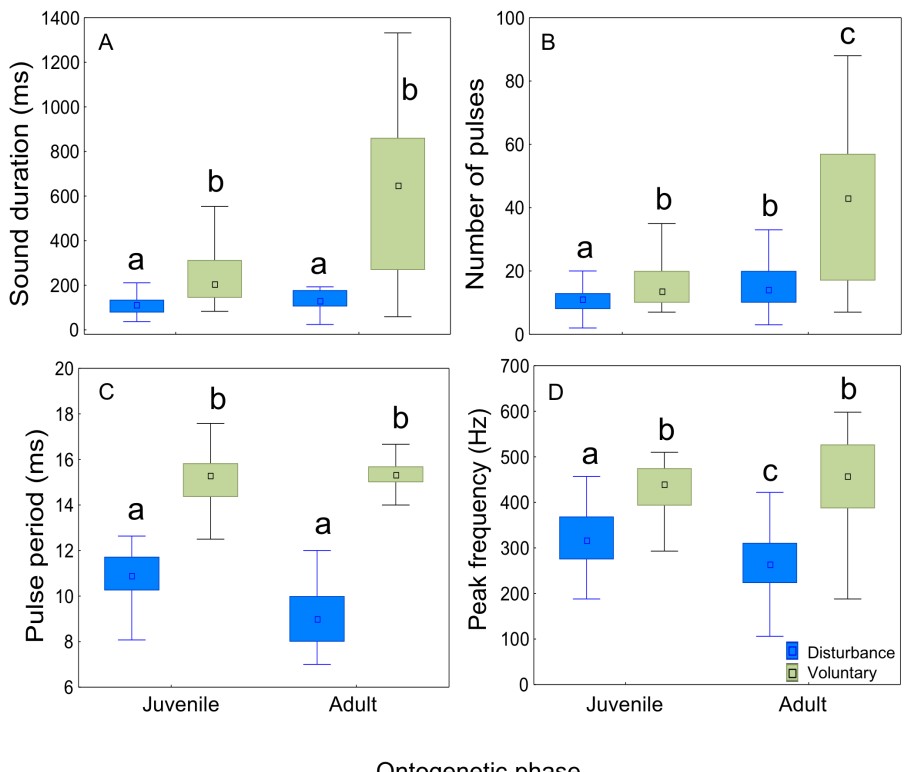

**Figure 5 Comparison of acoustic parameters of sounds emitted by juvenile and adult meagre according to context.** Fifty sounds were considered per ontogenetic class and context. The boxplots represent medians (central square), the 25th to 75th percentiles (boxes) and 5th and 95th (whiskers) for (A) sound duration (ms), (B) number of pulses, (C) pulse period (ms) and (D) peak frequency (Hz). Raw data is depicted. Different letters denote significant pairwise differences.

sounds recorded in captivity were pulse trains with $39 \pm 15$ pulses in adults and $16 \pm 6$ pulses in juveniles. The juvenile voluntary sounds were also typically formed by pulses with a single major cycle (Fig. 1), while the adult sounds (either in captivity or in the field) were typically formed by pulses with distinguishable multiple cycles. Additionally, the sound pulses recorded in the field appear to attenuate more rapidly than in the tank suggesting some reflections or tank resonance.

Plotting the several acoustic parameters against the number of pulses of sounds further illustrates the aforementioned differences between disturbance and voluntary sounds, in both juveniles and adults. Sound duration increased linearly with the number of pulses per sound both in voluntary and disturbance contexts for both ontogenetic groups, but in voluntary sounds the number of pulses increased more steeply than in disturbance sounds (ANCOVA; juveniles $F_{1,418} = 56,11$, $p < 0.001$; adults $F_{1,190} = 62.26$, $p < 0.001$; Figs. 6A, 6B). Sounds produced by adults were also longer and included a higher number of pulses than the ones produced by juveniles. In contrast to juveniles, the adults pulse number range was clearly wider in voluntary than in disturbance contexts (Figs. 6A, 6B; Table 3). Pulse period was longer and peak frequency higher in voluntary sounds of both

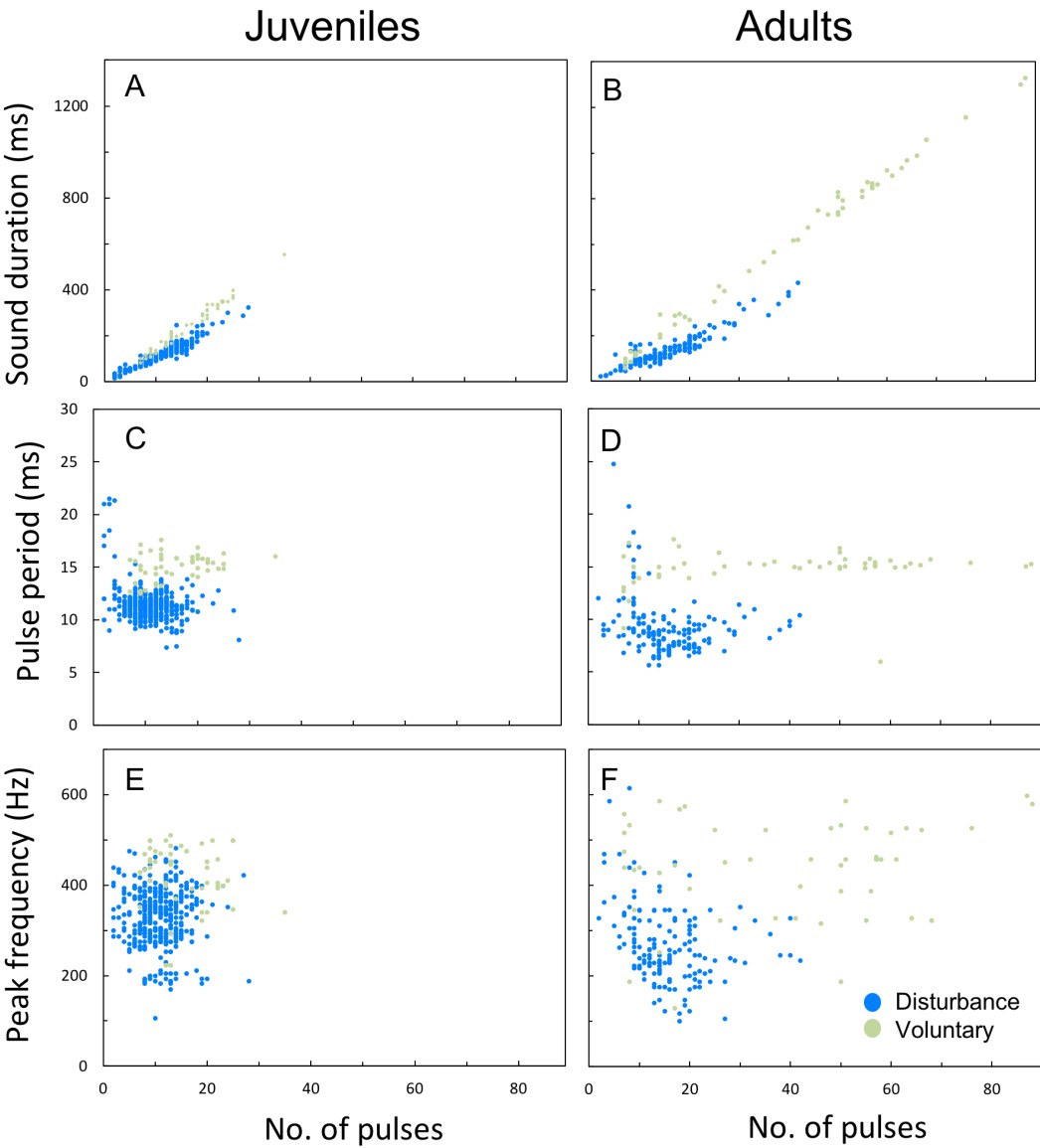

**Figure 6 Differences between disturbance and voluntary sounds observed in juveniles and in adults.** (A–F) Variation in sound duration, pulse period and peak frequency with the number of pulses for disturbance and voluntary calls, made by juvenile and adult meagre. Scatterplots were performed with 100 voluntary sounds (50 sounds per ontogenetic group) and 500 disturbance sounds (200 sounds for 10 adults; 400 sounds for 20 juveniles).

groups (Figs. 6C–6F). Pulse period was more irregular in disturbance than in voluntary sounds, especially in sounds with less than 5 pulses. The analysed voluntary sounds always presented more than 5 pulses. In both ontogenetic groups, peak frequencies of disturbance and voluntary sounds varied greatly with the number of pulses, showing no clear pattern (Fig. 6E, 6F).

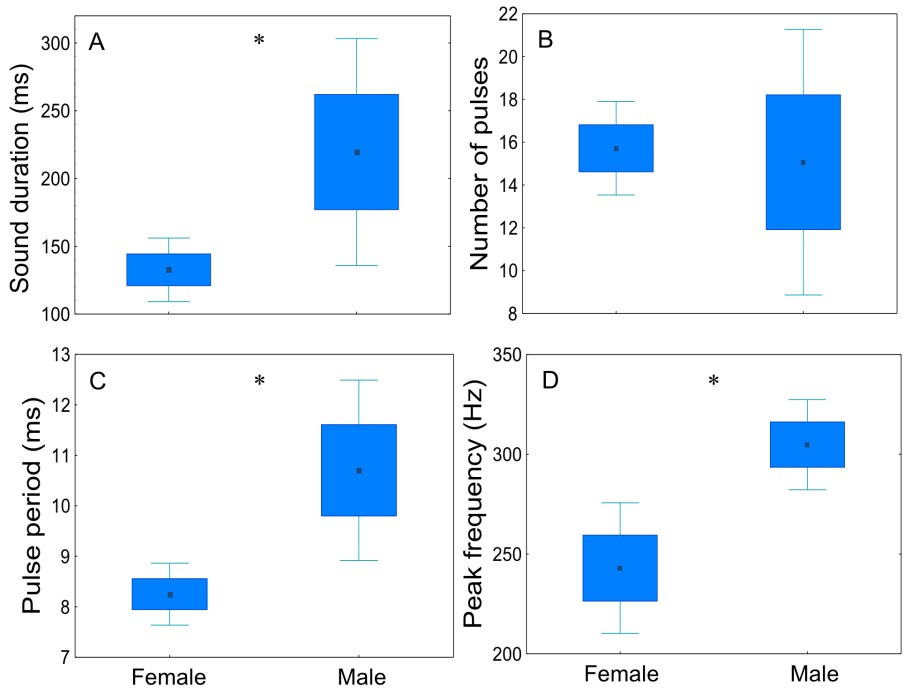

**Figure 7** **Comparison of sex-related differences in acoustic parameters of adult meagre's disturbance sounds.** Twenty sounds per fish (male $n = 4$, female $n = 6$) were considered. (A) sound duration (ms), (B) number of pulses, (C) pulse period (ms) and (D) peak frequency. The boxplots represent medians (central square), the 25th to 75th percentiles (boxes) and 5th and 95th (whiskers). Raw data is depicted. Parameters that are significantly different are indicated by * for $p < 0.05$ (Student's $t$-test).

### Sex variation of disturbance sounds

Sex-related differences of disturbance sounds characteristics were analysed for four males and six females (20 sounds per fish) (Fig. 7, Table 3). Disturbance sounds produced by males and females differed significantly in all acoustic parameters, with the exception of the number of pulses ($t = 0.23$; $p = 0.83$; d.f. $= 8$). Male sounds were longer than those emitted by females ($t = -2.35$; $p < 0.05$; d.f. $= 8$), had a longer pulse period ($t = -2.99$; $p < 0.05$; d.f. $= 8$) and a higher peak frequency ($t = -2.72$; $p < 0.05$; d.f. $= 8$).

### Field vs. captivity

The sounds registered in the field and in captivity consisted of sequences of pulses (not analysed) and grunts of variable durations and number of pulses. The field grunts showed similar sound duration ($t = -0.75$; $p < 0.001$; d.f. $= 118$), number of pulses ($t = 0.48$; $p < 0.001$; d.f. $= 118$) and peak frequency ($t = -0.90$; $p < 0.001$; d.f. $= 118$) to those emitted in captivity but differed in pulse period (ANCOVA; context $F_{1,117} = 61,53$, $p < 0.001$; water temperature $F_{1,117} = 30.20$, $p < 0.001$; Fig. 8, Table 3).

## DISCUSSION

Our knowledge on the detailed acoustic repertoire and variability of the sounds is lacking in most vocal fish species. The present dataset demonstrates for the first time

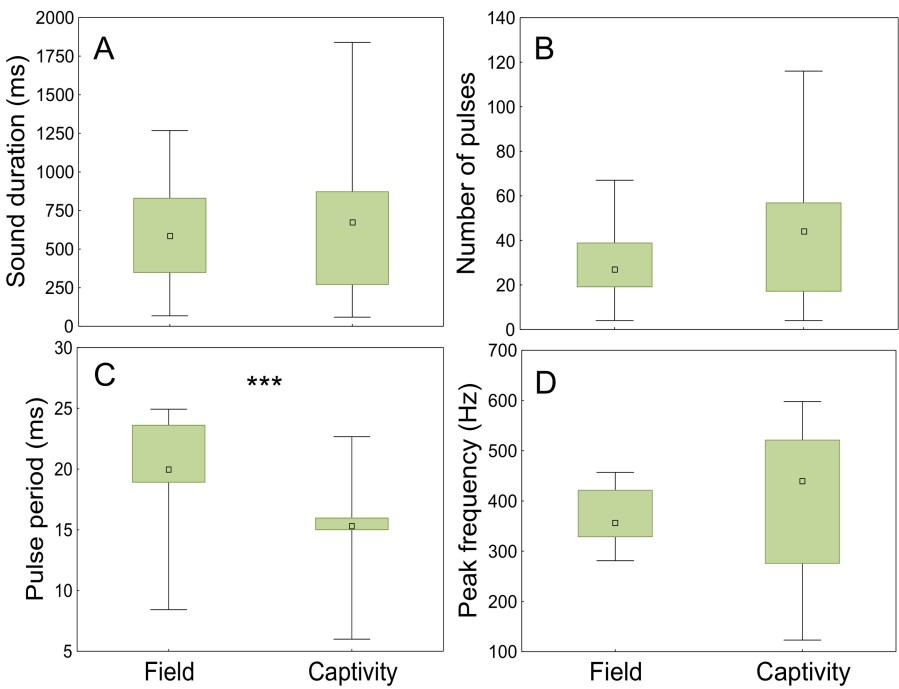

**Figure 8 Comparison of acoustic parameters of sounds emitted by meagre in the field and in captivity.** 50 sounds were considered for wild individuals and 70 sounds for captive fish. (A) Sound duration (ms), (B) number of pulses, (C) pulse period (ms) and (D) peak frequency (Hz). The boxplots represent medians (central square), the 25th to 75th percentiles (boxes) and 5th and 95th (whiskers). Raw data is depicted. Parameters that are significantly different are indicated by *** for $p < 0.001$ (ANCOVA).

that *A. regius* sounds varied with fish size, ontogenetic phase, sex, and context (disturbance and voluntary). Sound duration and number of pulses of disturbance sounds increased while pulse period, peak frequency and Q3 frequency decreased with increasing fish body size. Sexual dimorphism was significantly present in sound duration, pulse period and peak frequency of disturbance sounds, with males exhibiting higher values. Sounds produced in voluntary context showed significant differences between juveniles and adults, with adults exhibiting longer sounds with a higher number of pulses. Additionally, sounds produced by both adults and juveniles showed significant differences between disturbance and voluntary contexts, with disturbance sounds being of shorter duration, including fewer pulses, having a shorter pulse period and a lower peak frequency than voluntary sounds. Though grunts with less than five pulses are produced by this species in voluntary contexts (*Vieira et al., 2019*), in this study, these short sounds were not registered, likely because the sounds produced were connected with breeding in which sounds tend to be of long duration, or due to the short sampling effort. Finally, advertisement sounds recorded in Tagus estuary were similar to the ones registered in captivity although they differed in the pulse period, possibly associated with disparities in the environment and fish, such as the existence of higher temperature fluctuations in the field (*Ladich, 2018*) and different sizes or group composition of the sound emitters. Sounds registered in the field were from

unseen fish which were possibly larger than captive ones and also likely calling in larger aggregations.

## Ontogenetic and context variation of sounds

In this study, sounds emitted in two types of contexts were characterised for juveniles and adults *A. regius*: voluntary and disturbance. It is well-known that sciaenids are vocal species that produce sounds in courtship and disturbance contexts (*Connaughton, Taylor & Fine, 2000*; *Connaughton et al., 2002a*; *Lin, Mok & Huang, 2007*; *Tellechea, Fine & Norbis, 2017*). Though advertisement sounds produced during spawning aggregations have been characterised for *A. regius* (*Lagardère & Mariani, 2006*), disturbance sounds and their relation with size have never been studied for this species. As mentioned before, in *A. regius* (like in other sciaenids) sound production is elicited by the contraction of well-developed extrinsic sonic muscles that surround the swimbladder (*Tavolga, 1964*; *Ladich & Fine, 2006*). The waveform of both the disturbance and voluntary sounds of *A. regius* consisted of similar rapidly-damped pulses, suggesting that both sounds are produced by the same mechanism. Plus, contractions of the sonic muscles could be felt when handling the fish.

Disturbance sounds produced when juveniles (31–51 cm) and adults (69–102 cm) were handled varied significantly with fish size. According to *Connaughton, Taylor & Fine (2000)*, sound parameters such as pulse duration and peak frequency are determined by sonic muscle activity, rather than the resonant frequency of the swimbladder, which is thought to be strongly correlated with fish size. The increase of sound duration and decrease of peak and Q3 frequencies observed in this study support Connaughton and colleague's conclusion of a forced rather than a resonant response. Therefore, longer muscles, presumably with longer fibres, take longer to complete a twitch in larger fish resulting in longer pulses (presumably longer sounds) and also in lower sound frequencies. These results are in line with other studies on species belonging to the family Sciaenidae that demonstrated the same relation between these parameters and body size (*Connaughton, Taylor & Fine, 2000*; *Tellechea et al., 2010*) . However, whether *A. regius* is able to evaluate these differences to extract information still needs to be tested.

Two-way ANOVAs considering 50 sounds per ontogenetic phase and context showed that, when considering both disturbance and voluntary sounds, there was an increase in duration and number of pulses from juveniles to adults, but no variation was observed for the pulse period or peak frequency. These results were not surprising since the acoustic apparatus has been shown to increase throughout ontogeny in other sciaenids (*Hill, Fine & Musick, 1987*), likely increasing the ability to produce longer sounds with more pulses. Additionally, voluntary sounds produced by adults may be connected to breeding as sounds were recorded in the beginning of July, concurrent with the spawning period that characterizes this species (*Costa et al., 2008*). As such, it is expected that the sounds produced by mature, ready to spawn meagre are longer and include more pulses than the ones produced by immature juveniles. In contrast to what was observed for disturbance sounds, when considering sounds from both contexts, there was no decrease in peak frequency with fish size. This could be due to the high variability in this parameter arising from sounds presenting many harmonics with similar relative amplitude (Fig. 2).

The most marked variation in sound features was observed for the context of sound production. For both ontogenetic groups, significant differences between the two contexts were found for all parameters of sounds, with disturbance sounds being of shorter duration, including fewer pulses, and having a shorter pulse period and a lower peak frequency than voluntary sounds. In line with the observations made in this study, differences have been found between disturbance and voluntary sounds for other species in which agonistic sounds are brief and broadband (harsh) whereas sounds uttered in other contexts (submission, courtship) are frequently of longer duration and occasionally tonal (*Crawford, Hagedorn & Hopkins, 1986*; *Ladich & Tadler, 1988*; *Ladich et al., 1992*; *McKibben & Bass, 2001*). For example, the male midshipman (*Porichthys notatus*) produce tonal hums of long duration to attract pregnant females to the nest and short broadband grunts to defend the nest from potential intruders (*McKibben & Bass, 2001*). Physiologically, these differences are mainly related with differences in the contraction rate of sonic muscles (*Fine et al., 2001*). Furthermore, they are probably associated with differences in production rate and function of the two call types. According to *Lagardère & Mariani (2006)*, advertisement sounds of the meagre are mainly characterised by long grunts produced at a constant high rate for several tens of minutes. This implies higher costs of advertisement sounds, not only physiologically and metabolically (i.e., higher sonic muscle contraction rates for longer periods; (*Fine et al., 2001*), but also at an ecological level (e.g., attraction of predators). Consequently, it is likely that longer sounds and higher calling rates are used as direct signals of male quality and motivation for mate choice by females.

## Sex variation of disturbance sounds

The present dataset demonstrates for the first time that female meagre emit sounds and that their sounds differ from those of males. Both male and female *A. regius* emitted pulse trains (grunts) when handled. In most sciaenid species (e.g., *Cynoscion regalis* and *Sciaenops ocellatus*) only males possess sonic muscles (*Tower, 1908*; *Fish & Mowbray, 1970*; *Hill, Fine & Musick, 1987*). There are relatively few species in which these muscles are present in both sexes (e.g., *Micropogonias undulatus*, *Micropogonias furnieri*, *Pogonias cromis*, *Cynoscion guatucupa*, *Argyrosomus japonicus* and *A. regius*) (*Hill, Fine & Musick, 1987*; *Lagardère & Mariani, 2006*; *Ueng, Huang & Mok, 2007*; *Tellechea et al., 2010*; *Tellechea et al., 2011*; *Tellechea & Norbis, 2012*). Sexual dimorphism of the sound-producing apparatus has been observed in three species of the genus *Argyrosomus,* including the meagre (*Takemura, 1978*; *Lagardère & Mariani, 2006*; *Ueng, Huang & Mok, 2007*). Until now no information was available on the ability of females of this species to produce sound.

In this study, with the exception of the number of pulses, sexual dimorphism was significantly present in all of the studied acoustic parameters of sounds (sound duration, pulse period and peak frequency), with males exhibiting higher values. Hence, besides carrying information related to the size, dominant frequency and call duration could also be signals passing on information on the sex of the producer. There is a strong correlation between size and sex, with males usually having better developed sonic muscles associated to their larger size (*Allen, 1972*; *Ladich & Fine, 2006*). Male and female meagre used in this study did not differ significantly in total length. Females had a tendency to

be larger (mean TL of 87 cm for females vs. 82 cm for males) which could explain the higher peak frequency of their sounds. A similar result was observed for the closely-related species *A. japonicus* (*Ueng, Huang & Mok, 2007*). However, as in this experiment female gonadal tissue extraction was only succeeded for two individuals, female sex may have been incorrectly attribute to a non-sperm-producing male. Also sounds from more individuals should be recorded as the sample size was very small in the present study and results should be considered with caution.

Sound production by females could be associated with agonistic interactions as in other genus (e.g., *Cichlasoma centrarchus, Schleinzer, 1992*; *Trichopsis vittata, Ladich, 2007*). Whether females vocally take part in spawning is still unknown. Future experiments, in which advertisement sounds produced by *A. regius* are recorded for a single-sex group during the spawning season should provide answers to this question.

### Field vs. capticity

Advertisement grunts recorded in the Tagus estuary were similar to the ones recorded in captivity but differed significantly from those of captive individuals in pulse period. This may be explained by the diurnal temperature fluctuations observed in the field (ranging from 18 to 23 °C), which are highly influenced by tides, in contrast to more constant temperatures in captivity. Sound parameters associated with muscle contraction have shown to change with temperature (*Connaughton, Taylor & Fine, 2000*; *Connaughton, Fine & Taylor, 2002b*; *Ladich, 2018*). Moreover, water temperature appears to be the most important factor determining meagre migrations and reproduction (*FAO, 2005–2011b*). Additionally, the shorter pulse period of the sounds produced by captive individuals could be associated with different levels of motivation. While in the field the analysed sounds occurred outside the main chorus, in captivity individual sounds could be easily distinguished due to the smaller number of individuals. As such, it is likely that the latter were produced closer to spawning than the sounds analysed for wild fish.

### Concluding remarks

The variation in sounds documented in the present study indicates that *A. regius* vocalizations have the potential to carry information about species, size, sex, ontogeny, motivation, and other factors that may play a role in both voluntary and disturbance contexts. This variation in sound parameters may be valuable to fisheries biologists monitor natural populations of meagre with passive acoustic monitoring. For example, these differences could be used to detect meagre in the field and further attempt to ascertain ontogenetic phase (*Sprague & Luczkovich, 2001*), or to chart temporal and spatial patterns of meagre reproduction through the detection of spawning sounds as in other sciaenids (*Connaughton & Taylor, 1995*; *Luczkovich et al., 1999*; *Luczkovich et al., 2008*; *Rowell et al., 2017*). Further research to confirm differences between sex and ascertain the influence of other sources of variability such as water temperature could be valuable for such passive acoustic monitoring.

## ACKNOWLEDGEMENTS

We would like to thank Teresa Modesto for her contributions to this project.

### Funding

This study was funded by Fundação para a Ciência e a Tecnologia, Portugal (project PTDC/BIA-BMA/30517/2017 and project PTDC/BIA-BMA/29662/2017; SFRH/BD/115562/2016 to Manuel Vieira; and the strategic projects UID/MAR/04292/2019 granted to MARE and UID/BIA/00329/2019 granted to Centre for Ecology, Evolution and Environmental Changes). The funders had no role in study design, data collection and analysis, decision to publish, or preparation of the manuscript.

### Grant Disclosures

The following grant information was disclosed by the authors:
Fundação para a Ciência e a Tecnologia, Portugal (project PTDC/BIA-BMA/30517/2017 and project PTDC/BIA-BMA/29662/2017).
SFRH/BD/115562/2016 to Manuel Vieira.
Strategic project granted to MARE: UID/MAR/04292/2019.
Strategic project granted to Centre for Ecology, Evolution and Environmental Changes: UID/BIA/00329/2019.

### Competing Interests

The authors declare there are no competing interests.

### Author Contributions

- Beatriz P. Pereira conceived and designed the experiments, performed the experiments, analyzed the data, prepared figures and/or tables, authored or reviewed drafts of the paper, and approved the final draft.
- Manuel Vieira and Maria Clara P. Amorim conceived and designed the experiments, analyzed the data, prepared figures and/or tables, authored or reviewed drafts of the paper, and approved the final draft.
- Pedro Pousão-Ferreira, Ana Candeias-Mendes and Marisa Barata conceived and designed the experiments, authored or reviewed drafts of the paper, and approved the final draft.
- Paulo J. Fonseca conceived and designed the experiments, analyzed the data, authored or reviewed drafts of the paper, and approved the final draft.

### Animal Ethics

The following information was supplied relating to ethical approvals (i.e., approving body and any reference numbers):

The Portuguese National Authority for Animal Health provided full approval for EPPO to breed, use and supply aquatic animals for scientific experimental work (DGAV reference 0421/000/000/2018).

### Data Availability

Raw data is available as Supplementary Files.

## Supplemental Information

Supplemental information for this article can be found online at http://dx.doi.org/10.7717/peerj.8559#supplemental-information.

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
