# Peer review of "Sound production in the Meagre, Argyrosomus regius (Asso, 1801): intraspecific variability associated with size, sex and context"

_PeerJ, doi:10.7717/peerj.8559_

## Round 0.1 · original submission · Major Revisions

The reviewers have commented on your paper. They indicated that it is not acceptable for publication in its present form.

However, if you feel that you can suitably address the reviewers' comments (included), I invite you to revise and resubmit your manuscript.

·

Basic reporting

ok. Basically well-written, but some language glitches pop up.

Experimental design

ok

Validity of the findings

Fine except I would delete juveniles since the fish may not be juveniles. Lots of good information in this paper.

Additional comments

Pereira et al provide an interesting study of sound production in the Meagre, Argyrosomus regius using a development trajectory to compare sounds of captive juveniles, males and females and captive fish with wild ones. The paper is basically well-written and clear despite some minor errors that should have been caught. I do have some problems with the paper but believe they can be corrected.
First problem is the use of juveniles. I could grudgingly accept the 9 cm TL fish as a juvenile even with differentiated testes but think it is unlikely to be a fair descriptor for the 30-50 cm TL fish. First the authors provide no evidence of what constitutes a juvenile either with data or references to other studies. Secondly, I don’t think the term adds much. Their ontogenetic studies stand without using the word. Therefore either substantiate that these rather large fish are in fact juveniles or delete the term. I don’t think it adds anything to the paper.
I am also uncomfortable with the frequent use of the term gender, a squishy psychological term in place of sex. After all the fish are regular males and females. I can live with the term but see it as a mistake. Check with the American Psychological Association for more about gender.
The authors use the conventional term advertisement calls in the key words but repeatedly refer to social calls in the text and figures. Social calls immediately made me look for other kinds of interactions between individuals besides male courtship calls. Why be vague when there is a traditional term for what they describe.
L 43. I don’t see Radford et al in the refs?
L 52. 1986) ), ?
L 60. Tellechea et al (2010) from the J Exp Zool should be added to the ref string on black drum, particularly since the study compared male and female calls.
L 61. Acoustic rather than acoustics?
L 86. Sciaenids are not known to produce agonistic calls. I believe disturbance or distress calls would not typically be considered agonistic.
L 88. Allowing "recognition of" not allowing to recognize or.
L 92. Data provide.
L 110-113. Awkward and unclear. Rewrite.
L 114. Delete with.
L 115. "Were" not and not corresponded to.
L 120. Hard to believe that a 50 cm sciaenid is a juvenile. Delete or provide justification.
L 131. What is a sleeve and what is it made of? We need to know this to make sure it would not affect sound transmission.
L 138. I can guess, but what is an ichthyometer. Probably not necessary to give brand names for measuring TL and weight.
L 175. Were never fluid???
L 191. Delete The data set used in this study consisted of and add a “were” before obtained.
L 193. The hydrophone model was already mentioned. Once is enough unless there were different ones.
L 207. Again social sounds made me wonder if other reactions were taking place. Please stick with advertisement calls.
L 208-210. Awkward. Rewrite definition to make pulse period clearer. Perhaps use a second sentence.
L 214. Parameters were.
L 220. You state that all the sounds in this study were analyzed in one 5 hr period. Rewrite.
L 222. Which not witch.
L228. Delete and. It makes the sentence confusing.
L 240. Data considered mean values but sometimes all sounds were plotted (see Fig. 5). I think it would be better to stick with means less someone mention the P word (pseudoreplication).
L 248 level not levels?
L 253. Call not calls.
L 267. As far as I know, testicles are found in mammals not fishes. Use testes.
L 271. Re: all acoustic features. The authors don’t describe the acoustic waveform, which appears to be made of a single major cycle in the “juvenile” (Fig. 1) and multiple cycles in adults. This should be described and we need to know if the juvenile pulse is typical. Additionally the sound from the field appears to attenuate more rapidly than in the tank suggesting some reflections or tank resonance. Not a major concern, but it should be mentioned.
L 275. Delete second pulses.
L 277. Re: Fig 4. Not clear why we are going from regressions to box plots?
L 285-296. I am not convinced that this paragraph and Fig. 5 are providing new information.
L 304-305. The degrees of freedom, which should be 8, should be provided.
L 336. Pobably?
L 348. I asked Martin Connaughton who has watched weakfish producing sounds during courtship and he witnessed jockeying for position but no agonistic sounds?
L 365. Art Myrberg’s paper was on pomacentrids not sciaenids.
L 371. Not sure fighting ability is relevant here?
L 378. Number of pulses is determined by the cns. It should not be necessary to defend advertisement calls used in courtship here.
L 386. Energy of the remaining is unclear. Rewrite.
L 415. Would mention black drum here too.
L 420. Sounds like you may be repeating information already presented in the discussion.
L 422. Sentence is unclear: Usually, size and sex are extremely correlated due to the presence of sex size dimorphism.
L 425. Hill (1950)?
L 435. Swimbladder gas pressures typically match ambient pressure so there is a slack membrane, e.g. no relation to tension. Frequency of sounds is determined by time to contract and relax or the interval between contractions.
L 458. Worth to note???? Since you have the usual sounds (disturbance and advertisement calls) why speculate on what is missing?
L 474. Populations.
Michael L. Fine

·

Basic reporting

The article has clear and unambiguous, professional English used throughout. However, some improvements could be made:
Lines 110-113. I got lost here, especially the part on juveniles. Please re-write to improve the communication of what you want to say.
Lines 114-117. If the fish were in unisex tanks, why was there a sex ratio? Unisex should mean males and female were apart in separate tanks and, therefore, there would be no sex ratio. I did not understand something here. Please improve.
Line 267. Use testes rather than testicles.

The article completed the points on: Literature references, sufficient field background/context provided. Professional article structure, figures, tables. Raw data shared. Self-contained with relevant results.

Experimental design

Methods described with sufficient detail & information to replicate. I have some questions and observations.

I had trouble to understand the “n” used in the datasets that were analysed. The number of animals used is in the text as in the number of recordings per animal, but I found it hard to understand how these two “n” values were used and what the “n” was in the actual statistical analysis, i.e. the “n” per group in the tests. The authors should make this clear in the text and put the “n” in all tables and figures. The “n” should refer to the number of animals used. The authors should consult with a statistician. I think it is not correct to use repeated measures from the same animal as different data in an ANOVA or t-test. I suggest the authors use a repeated measures ANOVA that identifies the subject of the repeated measures and the treatments, origin, sex, context etc.

Line 257, an n of 4 males and 6 females does give an n=10. It is two groups with n=4 and n=6.

With the field data how were the authors sure that the sounds were from a single animal? Is it possible that the pulses came from more than one animal?
Can the size of the fish in the field data be estimated from the correlations? This could be interesting?

Figure 3. The correlation coefficients should be included. Some of the correlations look poor.

Line 238, Indicate that it is the “P” value.

The article completed the points on: Original primary research within Aims and Scope of the journal. Research question well defined, relevant & meaningful. It is stated how research fills an identified knowledge gap. Rigorous investigation performed to a high technical & ethical standard.

Validity of the findings

Impact and novelty not assessed. Negative/inconclusive results accepted. Meaningful replication encouraged where rationale & benefit to literature is clearly stated.

All underlying data have been provided; they are robust, statistically sound, & controlled. See comments in Experimental design section.

Conclusions are well stated, linked to original research question & limited to supporting results.

Speculation is welcome, but should be identified as such.

Additional comments

I was not convinced by the discussion on sound related to spawning. I do not think it is good practice to think that any sounds made during the spawning period, which is months, were related to spawning. Can the authors justify this assumption or remove it and text discussing this assumption? I think a closer relationship is required between the sounds and spawning. I would suggest that only sounds obtained at the time of spawning should be assumed to be sounds related to spawning. For example sounds recorded just before eggs are collect from a tank.

---

## Round 0.2 · Minor Revisions

I am pleased to confirm that your paper will be accepted for publication in PeerJ pending the final edits that are noted by Rev 1.

You have also emailed the PeerJ office to supply updated figures and legends, so this also gives you an opportunity to update those materials upon resubmission.

Thank you for submitting your work to this journal.

·

Basic reporting

Nice job.

Experimental design

Careful.

Validity of the findings

Yes.

Additional comments

I am pleased with the revisions to Peiria et al. although I still have a few minor comments.
L 44. I thought the ref to Myrberg was deleted since the sentence, at least as now written, is restricted to sciaenids, and he worked on pomacentrids.
L 74. Tellechea et al described male and female sounds in black drum and whitemouth croaker and should be mentioned here.
L 160. Posteriorly at the beginning of the sentence does not make sense.
L 224. Produced not poduced.
L 382. Naïve readers may not realize the significance of this conclusion, and I would therefore add “of a forced rather than a resonant response” after conclusion.
L 422, 446 legend for Fig. 7 and maybe other places. I thought you agreed to not use gender for sex?
L 597. Place author references starting with “R” after those starting with P.
Fig. 2 did not come though clearly in my copy.
Fig. 5. Incredibly minor, but the e in the legend for disturbance at the bottom right is partially up against the border.
Fig. 7. Male under the graphs needs to be moved to the right. I suspect the software did something unintended.

---

## Round 0.3 · accepted · Accept

Dear authors,

I am pleased to confirm that your paper has been accepted for publication in PeerJ. Thank you for submitting your work to this journal.